# Determination of USV’s Direction Using Satellite and Fluxgate Compasses and GNSS-RTK

**DOI:** 10.3390/s22207895

**Published:** 2022-10-17

**Authors:** Artur Makar

**Affiliations:** Department of Navigation and Hydrography, Polish Naval Academy, Smidowicza 69, 81-127 Gdynia, Poland; artur.makar@amw.gdynia.pl

**Keywords:** satellite compass, fluxgate compass, heading, course

## Abstract

The measurement of a mobile object’s movement direction is performed by means of various analogue and digital devices, including both autonomous and non-autonomous ones. They represent different measuring qualities, dimensions, weights and tolerance to ambient disturbances. They allow measuring the course of heading and course over ground (COG) in sea navigation. They are used for the determination of motion vectors on the water’s surface and with respect to the sea bed, in integrated systems, DP and autopilots. Results of dynamic tests of three heading meters: electronic and satellite compasses, and Global Navigation Satellite Systems (GNSS) determining COG are presented in this paper. The measurements were conducted in good measuring conditions, in an open upper hemisphere for satellite receivers and at no or minimal disturbances of the magnetic field. Sensors were mounted on an unmanned survey vessel (USV) that was moving straight, performing quick turns and circulations. Each of them has some limitations with respect to its use in the water area in which a hydrographic sounding is to be performed; attention was paid to the possibility of using a given compass on board a small autonomous ship navigating automatically.

## 1. Introduction

The determination of the position of an object moving in a given area is the first aim of navigation and is necessary in order to reach a given point in a safe way, with predictable accuracy and in an expected amount of time. This task is executed in an autonomous manner with the use of various onboard measuring devices and systems. Magnetic compasses and gyrocompasses were the first instruments used for this reason. These devices facilitate the determination of the object’s position based on measurements of the Earth’s magnetic field strength, positions of astronomical objects and permanent characteristics of land objects. Due to the diversity of applicable devices, navigation may be divided into the following sections: (1) astronomical navigation, consisting of observation of the astronomical objects; (2) terrestrial navigation, consisting of observation of beacons and other characteristic objects present on the coast; (3) dead reckoning, consisting of the approximate determination of the ship’s position based on her last known and measured position, the direction of her heading and speed; (4) pilot navigation, which takes into considerations the beacons present in the given water areas, used close to seaports, on fairways to the ports and in other marked places difficult to navigate in; (5) radio-navigation, which is based on radio signals sent via transmitters; (6) satellite navigation, which uses radio signals transmitted by fake Earth satellites; and (7) inertial navigation.

Despite positioning, the determination of direction and measurement of speed are indispensable measuring processes during navigation. Gade [1] defined seven methods of plotting a course. They were used in the construction of various devices, both autonomous and non-autonomous, of different accuracy classes, with different possibilities of cooperation with other devices and various resistances to the influence of ambient factors. Apart from the magnetic and standard compasses, gyro-compasses [2,3,4,5,6,7] are used on big ships due to their significant dimensions and weights. A satellite compass (of one system) operates correctly in good observation conditions [8,9,10,11,12,13,14], and it is useless in seaport areas with high buildings close to the ships.

Both headings as well as COG are used in hydrographic systems for registration needs, digital presentations for the system operator and helmsman, and graphical presentations of the motion vector. On big sounding vessels equipped with gyro-compasses, a gyro-compass heading is used to present the motion vector on the water surface. It is also possible to use a satellite compass in good observation conditions, i.e., on open water.

The miniaturisation of vessels used in hydrography [15,16,17,18,19,20,21,22,23,24,25], especially unmanned ones (USVs), and of measuring devices, in particular, the acoustic ones, including single beam echosounders (SBES) and multibeam echosounders (MBES), provides the possibility of their usage in limited water areas, especially in seaports where the ground conditions restrict the application of satellite devices and systems and where disturbance of the electromagnetic field caused by ferromagnetic objects is present. Compasses installed on USV, being an integral part of the autonomous measuring system, are not resistant to such disturbances, which results in no possibility for the sounding vessel to head along the sounding profiles in an automatic mode. Therefore, it is necessary to apply a manual mode or another device to plot the course of a mobile object to operate the USV in a precise way in the automatic mode [26,27,28,29,30,31,32,33] in a quay, among the stand places of yachts in marinas and berthed big ships. Advanced equipment such as MBES, LiDAR, RADAR, cameras, and side scan sonars (SSS) are used in bigger USVs and sounding vessels equipped with GPS compass for heading determination. Small USVs use basic hydrographic sensors: SBES for depth measurement and GNSS for positioning. For automatic navigation, they use an internal electronic compass. The article presents the results of heading determination using sensors, which can be mounted on board small USVs.

The measurements were done in the Gdynia marina neighbouring a municipal beach (Figure 1). It is protected by a breakwater, which ensured no impact of hydro-meteorological conditions on the USV.

Three devices enabling the course of the USV to be plotted were used for this research: a fluxgate compass, being an electromagnetic instrument, and two satellite receivers: one-antenna multi-GNSS for COG determination and a two-antenna one-system satellite compass. They represent various qualities with respect to ambient resistance. On the one hand, the fluxgate should determine the course in a water area of difficult GNSS observation conditions, but it is sensitive to the magnetic field influence. On the other hand, satellite devices are resistant to external conditions but in the open upper hemisphere enabling receipt of signals from the satellites.

The composition of the paper is as follows: Section 2 presents the measuring vessel and sensors used to plot her course. Section 3 provides the results of static and kinematic measurements done in a seaport area on long straight profiles, short profiles and on returns. Conclusions close the article.

## 2. Materials and Methods

### 2.1. Measured Parameters and Relationships between Them

In a measuring system (Figure 2), each of the sensors determines the direction between the northern part of the meridian passing the given sensor or arrangement of the sensors and the beginning of the coordinates polar system related to the sensor. As a reference, an arrangement of the satellite compass, for which (at the base distance between antennas being 120 cm) it may be assumed that it coincides with the USV’s centre line, has been accepted.

The fluxgate compass was installed approximately towards the centre line, and then, after the execution of static measurements, corrections were done in an analytic way. It is also possible to compensate it mechanically within a range of several degrees. The COG is not subject to any initial measurement nor calibration due to being a parameter determined based on the coordinates of the position. However, parameters measured by the fluxgate and satellite compasses are connected with the sounding vessel’s hull, and COG is determined on the basis of the GNSS antenna’s positions’ coordinates. There is a possible difference between COG and parameters measured by two other sensors, which can be observed in Figure 2. Generally, the hydro-meteorological conditions of wind and wave motion do not influence the measuring objects and determined parameters. The discrepancy between HDT, HDM, and COG is a result of the difference between parameters related to water and the ground.

The satellite compass, as a GNSS sensor, works in an open area, so it can be impossible to use it in a restricted area with high port infrastructure, close to high buildings and high moored vessels. Measurements are provided. The GNSS receiver in RTK/RTN mode can be used in the GSM range (operational zone), but the article is dedicated to hydrographic surveys in coastal areas using small USVs with the availability of RTK/RTN corrections. It is a partial answer to the next question about measurements in the open sea.

### 2.2. USV and Measurement Sensors

An OceanAlpha SL20 USV (Figure 3) is a hydrographic vessel equipped with an SBES Echologger and an internal GPS receiver, which is usually replaced with a geodetic GNSS receiver. It is powered by two (water-jet propulsion) engines and has no rudder. During bathymetric surveys, it moves with a velocity of 2–5 kn, which enables both line keeping and steering. The data are transmitted via a radio link at a frequency of 2.4 GHz. A manipulator that allows two engines to be controlled separately is used for manual steering. Autopilot, using a course from the internal electronic compass, ensures automatic navigation (i.e., the guidance of USV along planned lines).

A two-system (GPS and GLONASS—GLObalnaja NAwigacionnaja Sputnikowaja Sistiema) Leica Viva GS15 [34,35] receiver was used as an external receiver to determine the position’s coordinates in real-time network (RTN) mode, with a horizontal accuracy not worse than 1 cm thanks to the corrections being received from the SmartNet network.

Antennas of the satellite compass Novatel PwrPak 7 [36,37] were mounted on a rigid measuring system at a distance of 120 cm. The receiver of the compass enables the cooperation with other devices and dedicated software via a serial port, a RJ45-TCP/IP wire link and a cordless network, with the possible registration of the observation data internally. The RJ45 cordless link was used in the tested measuring system of the internal USV network. The pitch angle is an additional parameter determined by the satellite compass [38,39,40,41,42,43,44,45,46,47,48,49].

Plotting the compass course is a basic task executed by an electronic compass. The measured resultant vector of the magnetic field is corrected automatically (auto-compensation) by the deviation value, so it equals the magnetic course value [50,51,52,53,54,55]. An INI-200 ATC compass was used in the measurements.

Table 1 presents the basic parameters of the USV and sensors.

The integration of sensors was made based on an internal network enabling the application of an internal GPS receiver, compass, echosounder and external GNSS receiver operating in geodetic RTK (real-time kinematic)/RTN variant. Additional sensors were connected to the network: the Novatel PwrPak 7 satellite compass, which was connected directly by means of the TCP/IP protocol, and the fluxgate electronic compass, which was connected with the use of an RS232—TCP/IP converter WRT610.

Table 2 presents messages used to register the heading/course:

An example of recorded messages is shown below.
GNSS   $GPGGA,082319.00,5431.0669492,N,01833.0649214,E,4,08,1.22,−4.0236,M,33.6965,M,2.0,0871*51$GPGLL,5431.0669492,N,01833.0649214,E,082319.00,A,D*6D$GPGSA,M,3,32,06,19,22,25,17,24,12,,,,,2.23,1.22,1.87*09$GPGSV,3,1,09,32,27,311,41,06,23,097,40,19,42,060,46,22,14,326,40*71$GPGSV,3,2,09,25,27,258,39,17,25,047,42,24,68,166,49,12,70,253,48*7A$GPGSV,3,3,09,02,10,139,42*4E$GPVTG,187.581,T,,M,1.2603,N,2.3341,K,D*0B $HCHDM,241.9,M*27Fluxgate   $GPHDT,138.0153,T*08Satellite compass   $HCHDM,241.9,M*27

HYPACK (HYPACK, Middletown, CT, USA) software was used for the recording of geospatial data during a bathymetric sounding and heading/course during measurements.

## 3. Results

### 3.1. Static Measurements—Fluxgate and Satellite Compass

Prior to the execution of the static measurements by means of the fluxgate electronic compass, its calibration was performed by means of full circulation to the right, with a double crossing of the North direction. It was fixed into the measuring system rigidly connected with the base (a two-antenna system) of the satellite compass. The satellite compass did not require calibration. The measuring session lasted 180 min. Figure 4 shows the course plotted by the satellite compass (**a**) and measured by the fluxgate (**b**).

On the basis of the statistical analysis, the mean value of the heading HDG (HDT and HDM) is as follows:(1)$$\bar{HDG}=\frac{\sum^{\;}HDG}{n}=\{\begin{array}{c} 293.44{}^{\circ}\\ 290.82{}^{\circ} \end{array}\begin{array}{c} for\;Novatel\\ for\;fluxgate \end{array}$$
and the standard deviation is:(2)$$\sigma_{HDG}=\sqrt{\frac{\sum^{\;}{\left(\Delta HDG\right)}^{2}}{n}}=\{\begin{array}{c} 0.0658{}^{\circ}\\ 2.835{}^{\circ} \end{array}\begin{array}{c} for\;Novatel\\ for\;fluxgate \end{array}\;.$$
where *n* denotes the number of recorder headings. Because of different standard deviations, two parts of Figure 4 were presented in Figure 5: the first 10 s of static measurements (**a**) and a period between 5.5 and 6.5 s (**b**).

Next, the correction of the fluxgate heading has been calculated as the difference between satellite and fluxgate headings. The correction of 2.6° has been added to each of the measured headings using the fluxgate compass. 

### 3.2. Linear Trajectory—Long Profiles

In the first part of dynamic measurements, 50 m long linear profiles were planned with a distance of 2 m between them [56]. The automatic mode of USV steering allowed us to keep the profile (line keeping) with minimum cross-track error (XTE) (Figure 6) as a distance between the USV position and the profile. 

Figure 7 presents satellite compass and fluxgate headings and COG during a 500 s survey of USV on 9 linear profiles.

The first 200 s of the survey divided into four parts is shown in Figure 8.

### 3.3. Quick Heading’s Changes on Reciprocal One

For measurements of quick heading changes, short 12 m profiles were planned with a 5 m distance between them [56]. The goal of measurements was to record and analyse headings while turning the USV 180° right and left. As distinct from the navigation on long profiles, when the USV’s speed is almost constant, in short profiles, the speed varies. The lowest speed occurs during turning on another, closer profile. The USV’s position should be at the beginning of the profile with the heading as the profile’s bearing. The rate of turn (ROT) is low, and the 1 Hz frequency of message transmission (fluxgate) enables the observation (recording) of low heading changes. Figure 9 shows the trajectory of the USV during measurements on short, parallel profiles.

Figure 10 presents satellite compass and fluxgate headings and COG during a 270 s survey of USV on 13 short profiles.

For observation differences of measured headings/COG, Figure 11 presents the results of measurements of two first profiles and two first turnings.

### 3.4. Circulations

The goal of measurements during circulation was to record and analyse the constant heading/COG as a result of long-term USV turning. The rotation has been obtained manually by working only one propulsion: the left engine for rotation to the right and the right engine for rotation to the left. Figure 12 presents the trajectory of USV during circulations: left (blue line) and right (red line).

Figure 13 presents headings and COG during 90 s of rotation of the USV to the right (a) and 120 s of rotation to the left. During measurements, the USV rotated four times right and five times left.

Figure 14 presents headings/COG during two rotations right (**a**), (**b**) and left (**c**), (**d**). They took about c.a 24 s (23 s–25 s) with ROT ≈ 15°/s.

## 4. Discussion

(1) While executing the static measurements, a mean value of the heading measured by the fluxgate and satellite compasses was determined; it was indispensable to determine correction for the electronic compass heading with respect to the satellite compass. A standard deviation was determined of 0.07° for the satellite compass and 2.84° for the fluxgate.

(2) The best accuracy of the satellite compass was assumed, and the assumption was also that it was a reference sensor in good (sufficient) conditions for the satellites to determine the heading, despite cooperation with only one GPS satellite system in the SBAS variant. The delivered software monitors the accuracy of determining heading and yaw parameters and of the signal reflection from buildings.

(3) Indications of the fluxgate electronic compass vary from indications of the satellite compass in a serious way (even by 50°) during return and, afterwards, what is important in navigating the measuring vessel along the sounding lines. In dynamic conditions, even on board such a vessel as a USV that moves very slowly, high accuracy of the heading measurement at return with high frequency and low inertia is important. Figure 15 presents heading differences between the satellite compass and fluxgate.

(4) COG does not differ from the heading determined by the satellite compass in motion in a special way (difference between indications of the satellite compass and COG), even at low, and for the vessels, velocities if there are no impacts of hydro-meteorological factors there. Figure 16 presents differences between satellite compass heading and COG.

## 5. Conclusions

Three course meters (out of many different compasses used in navigation) were applied: two compasses and a GNSS receiver determining COG. Static measurements of the compasses were necessary to determine corrections for the electronic compass with respect to the satellite compass. Both sensors represent high-angle stability and are state-of-the-art for their ages (please note that the fluxgate has not been manufactured since 2002). The declared 0.1° accuracy of the course measurement is sufficient for the USV navigation with a measurement frequency of 10 Hz.

The fluxgate compass does not meet the requirements of up-to-date dynamic measurements on board a ship operating in low and very low disturbances of the magnetic field due to high and increasing errors of the course measurements despite the deviation auto-compensation.

COG is an important parameter of a moving object, taking into consideration the impact of external factors on a vessel. At their lack, vectors of motion on the water surface coincide, and COG may be used to draw the vessel’s symbol on an electronic map of the hydrographic system in real-time in order to operate manually along sounding lines. In addition, coordinates of the position and the course based on COG are profiled thanks to the 1-3-step Kalman filter (FK). Due to the influence of hydro-meteorological factors on the vessel, COG is not applicable as the autopilot’s sensor to operate the ship along the lines automatically. The discrepancy between COG and the heading also occurs during the execution of return onto the next line (onto a counter heading) due to the occurrence of inertia.

## Figures and Tables

**Figure 1 sensors-22-07895-f001:**
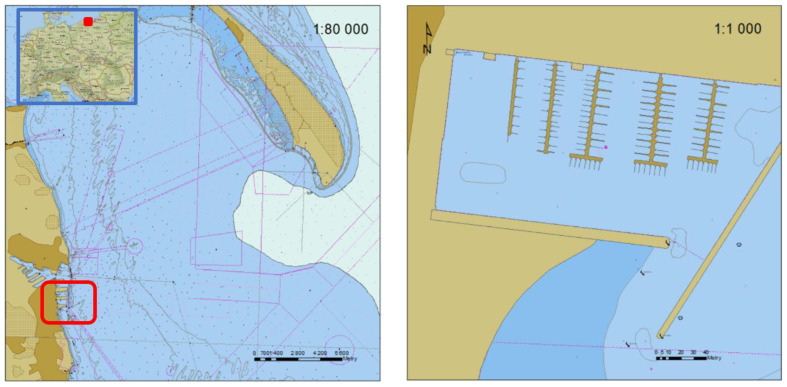
The restricted area of USV surveys and heading/course measurements.

**Figure 2 sensors-22-07895-f002:**
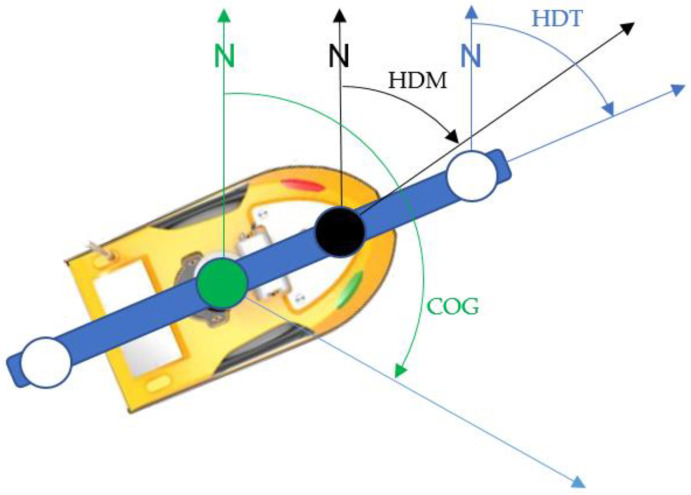
Relationships between headings and COG.

**Figure 3 sensors-22-07895-f003:**
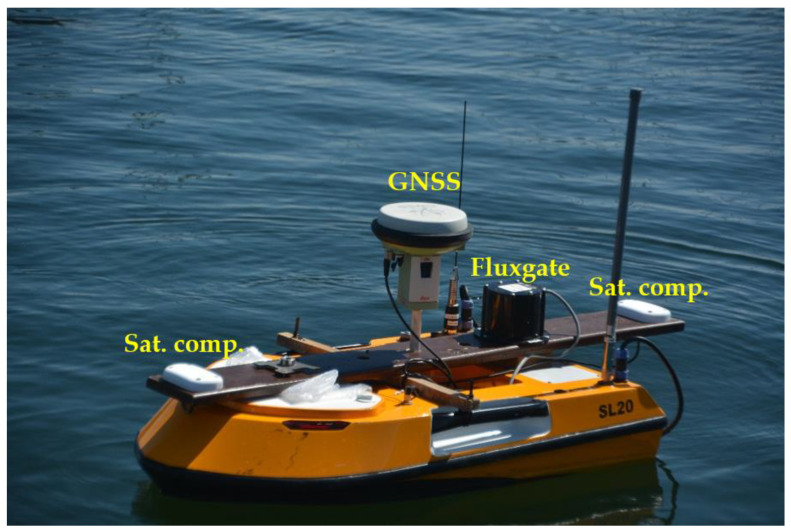
OceanAlpha USV SL20 during multi-sensor heading and course measurements.

**Figure 4 sensors-22-07895-f004:**
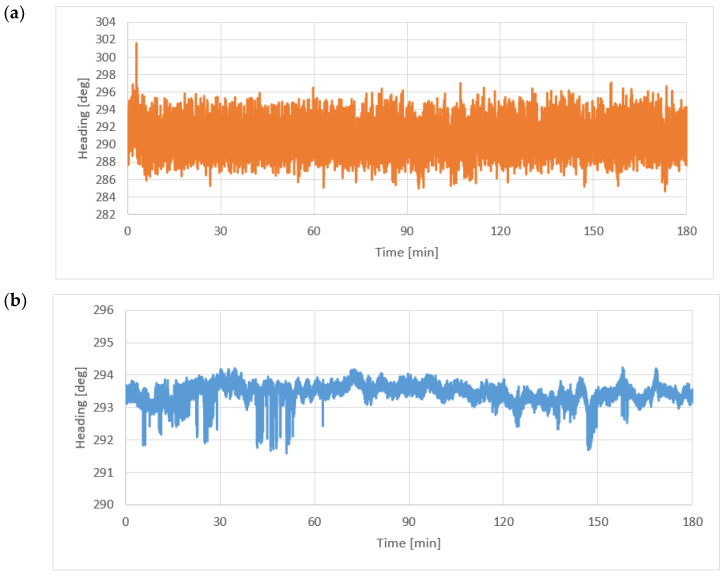
Static measurements of the course using a satellite compass (**a**) and a fluxgate (**b**).

**Figure 5 sensors-22-07895-f005:**
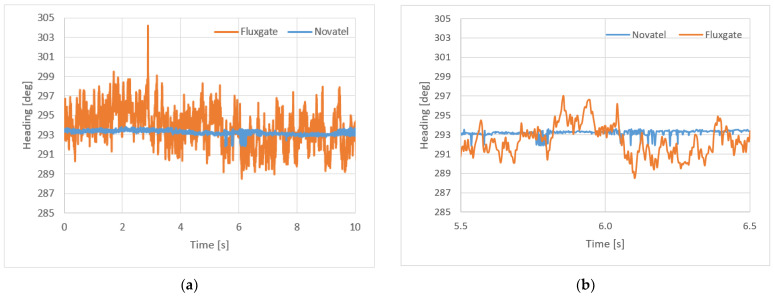
First 10 s of static measurements (**a**) and a period between 5.5 and 6.5 s (**b**).

**Figure 6 sensors-22-07895-f006:**
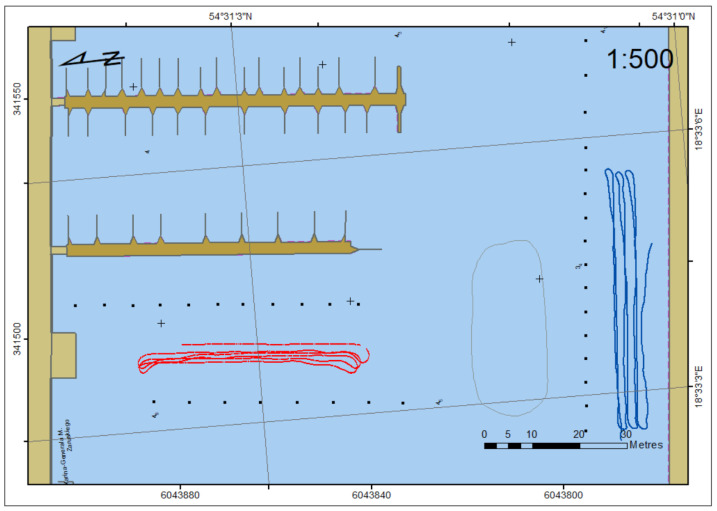
Trajectory of USV during measurements on long, linear profiles.

**Figure 7 sensors-22-07895-f007:**
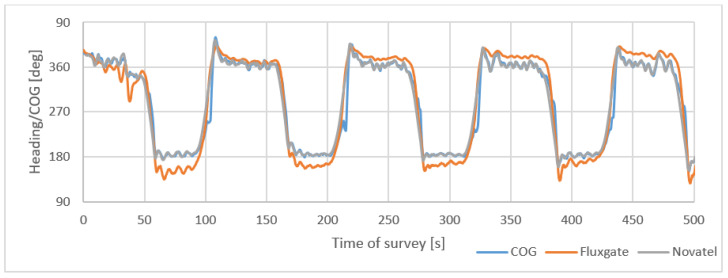
Satellite compass and fluxgate headings and COG during 500 s survey on linear profiles.

**Figure 8 sensors-22-07895-f008:**
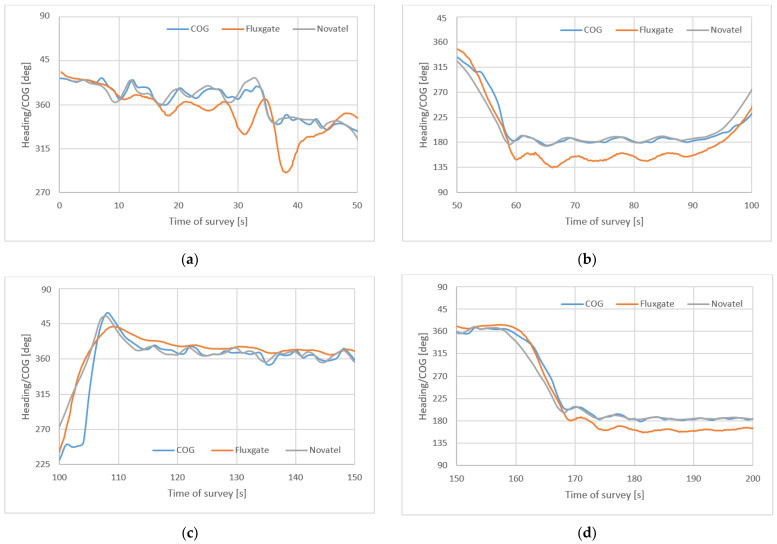
A part of the USV survey on the first (**a**), second (**b**), third (**c**) and fourth (**d**) profiles.

**Figure 9 sensors-22-07895-f009:**
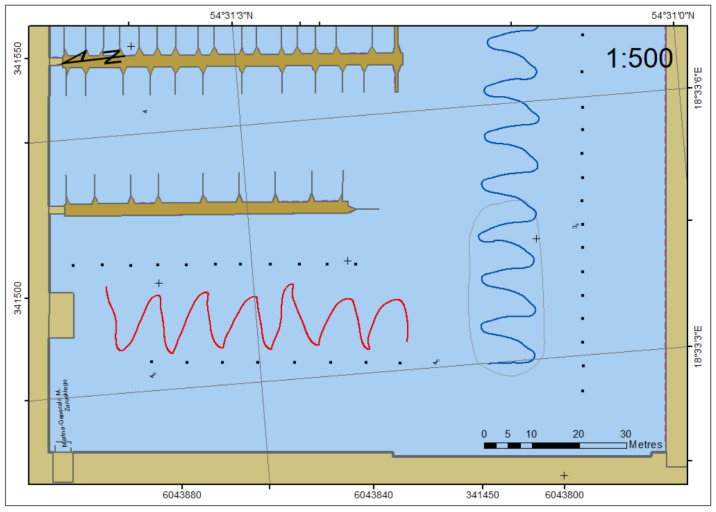
Trajectory of USV during measurements on short profiles with quick heading changes.

**Figure 10 sensors-22-07895-f010:**
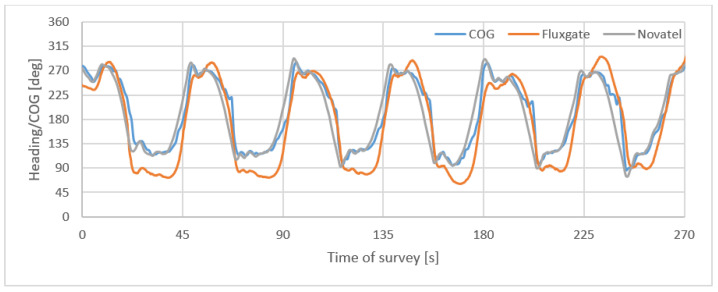
Satellite compass and fluxgate headings and COG during 270 s survey on short profiles with quick heading changes.

**Figure 11 sensors-22-07895-f011:**
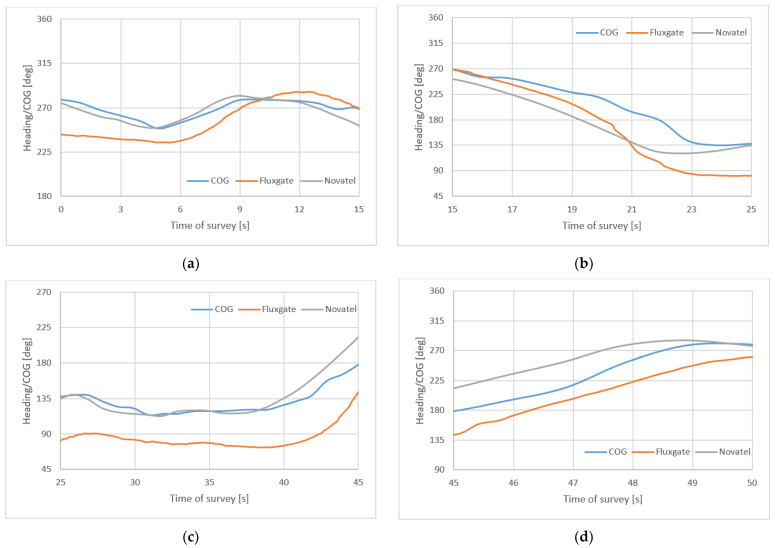
Headings and COG recordings of two first profiles (**a**,**c**) and two first turnings (**b**,**d**).

**Figure 12 sensors-22-07895-f012:**
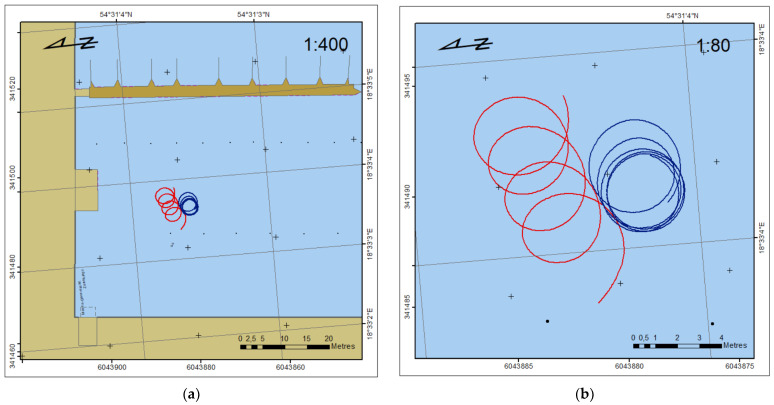
Trajectory of USV during circulations in two scales: small (**a**) and large (**b**).

**Figure 13 sensors-22-07895-f013:**
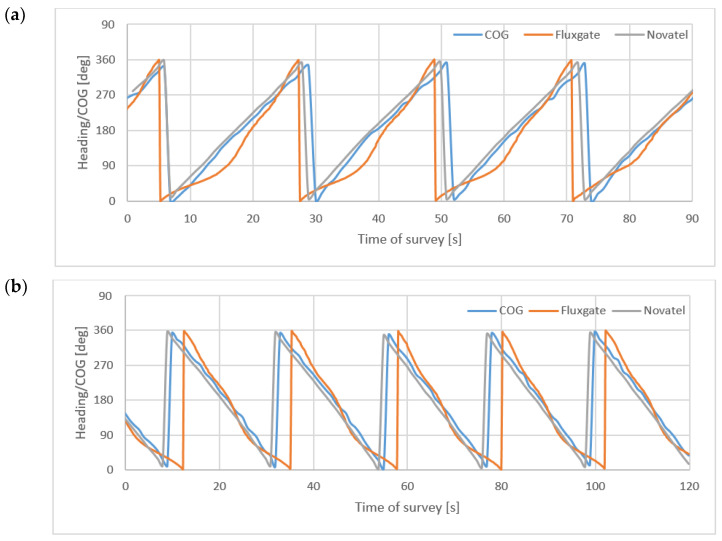
Headings and COG during 90 s rotation of the USV to the right (**a**) and 120 s rotation to the left (**b**).

**Figure 14 sensors-22-07895-f014:**
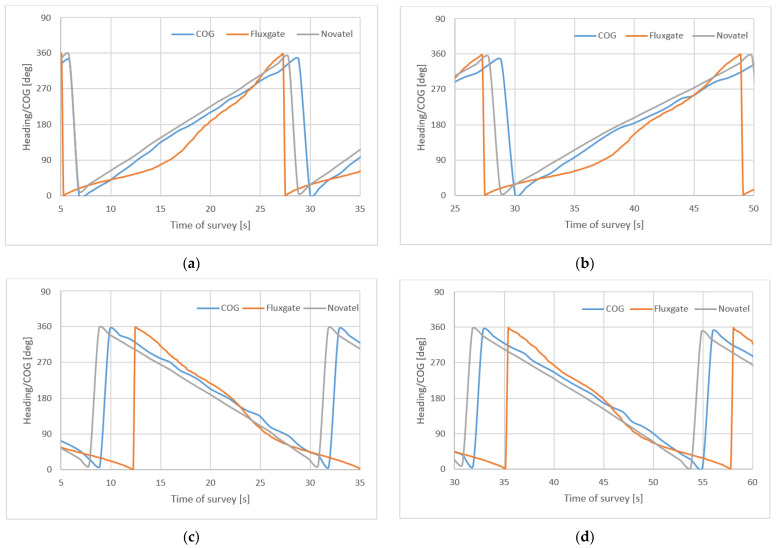
Headings/COG during two rotations right (**a**,**b**) and left (**c**,**d**).

**Figure 15 sensors-22-07895-f015:**
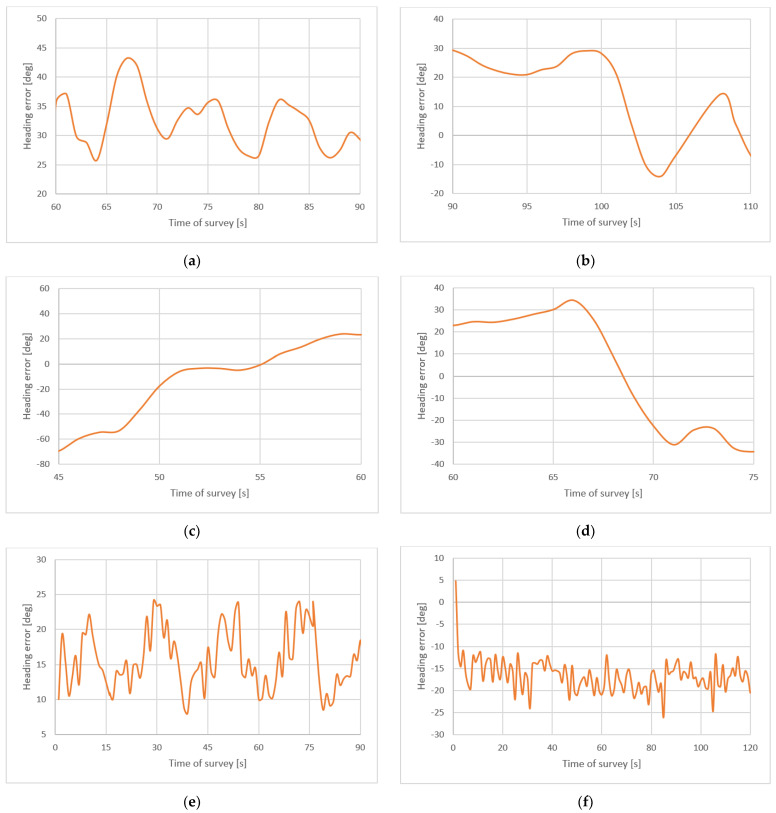
Heading differences between satellite compass and fluxgate during surveys: long profile (**a**) and turning (**b**), short profile (**c**), quick heading changes (**d**), and circulations right (**e**) and left (**f**).

**Figure 16 sensors-22-07895-f016:**
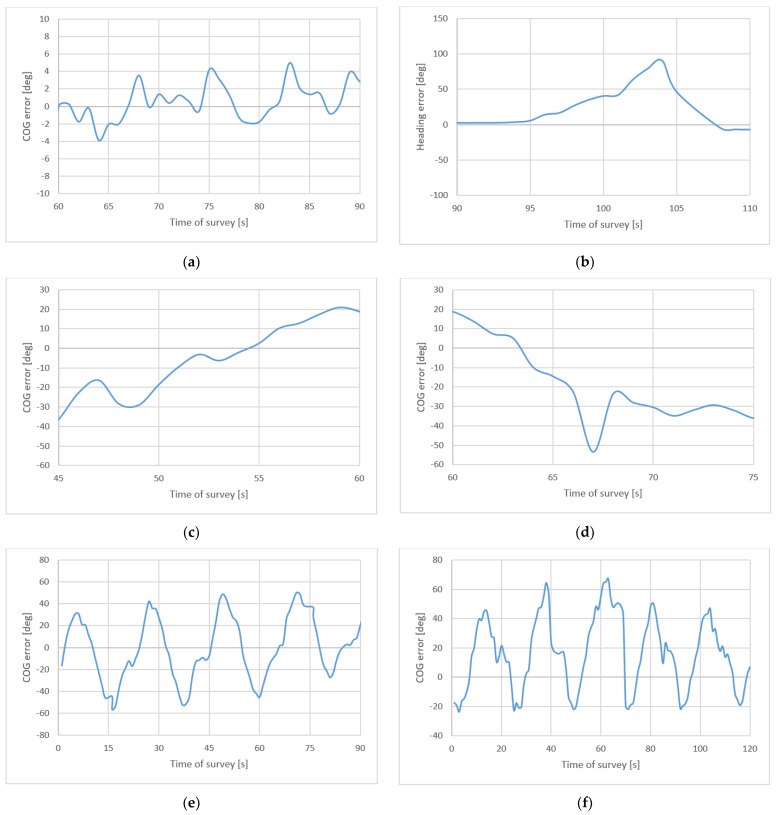
Differences between satellite compass heading and COG during surveys: long profile (**a**) and turning (**b**), short profile (**c**), quick heading changes (**d**), and circulations: right (**e**) and left (**f**).

**Table 1 sensors-22-07895-t001:** Basic parameters of USV and sensors.

Device	Photograph	Parameter	Value	
SL20	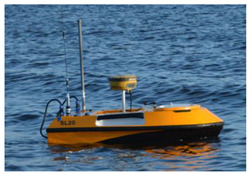	Hull material	Carbon fiber	
		Dimension	105 cm × 55 cm × 35 cm	
		Weight	17 kg	
		Draft	15 cm	
		Propulsion	Water-jet propulsion	
		Survey speed	2–5 kn (1–2.5 m/s)	
		Max speed	10 kn (5 m/s)	
		Positioning (standard—not used)	U-blox LEA-6 series	
		Positioning (used in maneuvering)	Leica Viva GS15	
		Heading	Honeywell HMC6343	
		Echosounder	Echologger series SBES	
Novatel PwrPak 7	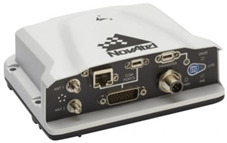	Signals Tracked Primary Antenna	GPS (L1 C/A, L1C, L2C, L2P, L5) GLONASS (L1 C/A, L2 C/A, L2P, L3) BeiDou (B1I, B1C, B2I, B2a, B2b) Galileo (E1, E5 AltBOC, E5a, E5b) NavIC (IRNSS) (L5) QZSS (L1 C/A, L1C, L1S, L2C, L5) SBAS (L1, L5) L-Band (Up to 5 channels)	
		Code measurement precision	GPS GLONASS BeiDou Galileo	4–8 cm 8 cm 4 cm 3 cm
		Velocity accuracy	<0.03 m/s RMS	
		ALIGN heading accuracy	Baseline = 2 m 0.08 degrees Baseline = 4 m 0.05 degrees	
Fluxgate INI-200 ATC	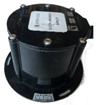	Accuracy	0.2°	
		Maximum operating magnetic inclination	85°	
		Gimbal operating range Heel (roll) angle Pitch angle	±45° ±45°	
GNSS Leica Viva GS15	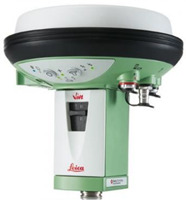	GNSS technology	Advanced four constellation tracking	
		Number of channels	120 (up to 60 satellites simultaneously on two frequencies)/500 + 1	
		Signal tracking	GPS (L1, L2, L2C, L5), Glonass (L1, L2), BeiDou (B1, B2), Galileo (E1, E5a, E5b, Alt-BOC) QZSS (L1, L2, L5), SBAS (WAAS, EGNOS, MSAS, CAGAN)	
		Code differential DGPS/RTCM	Typically 25 cm	
		Real-time kinematic Single baseline (<30 km) Network RTK	Hz 8 mm + 1 ppm/V 15 mm + 1 ppm Hz 8 mm + 0.5 ppm/V 15 mm + 0.5 ppm	
		Post-processing Static (phase) with long observations Static and rapid static (phase)	Hz 3 mm + 0.1 ppm/V 3.5 mm + 0.4 ppm Hz 3 mm + 0.5 ppm/V 5 mm + 0.5 ppm	

**Table 2 sensors-22-07895-t002:** Messages transmitted by sensors.

Device	Message	Parameter	Frequency [Hz]
GNSS	$GPGGA, $GPGLL $GPVTG $GPGSA, $GPGSV	Position coordinates COG, SOG	10
Satellite compass	$GPHDT	Heading	10
Fluxgate compass	$HCHDM	Heading	1

## Data Availability

Not applicable.

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
