# Peer review of "Determination of USV’s Direction Using Satellite and Fluxgate Compasses and GNSS-RTK"

_sensors, 2022, doi:10.3390/s22207895_

Round 1
Reviewer 1 Report
The objective of the paper is the comparison of fluxgate and satellite compasses, and Global Navigation Satellite Systems (GNSS) determining COG of USV. - It is needed to elaborate more in the added value of the paper related with existing literature describing the weaknesses identified - Specify why particular devices are selected for performing trials related with similar existing in the market - In the methodology of comparison measurement describe the reason for selection the protocol and if they have consulted some similar measurements - Erase/translate the first paragraph of 3.4 Circulation section which is in Polish - Describe what are the limitations of your measurement - Elaborate more in hydro-meteorological factor on the vessel and potential of measurement in another area (open sea???) - Generalize your conclusion and propose future research
Author Response
According to Reviewers comments and Editor’s suggestion, that the authors have not adequately
proved the novelty of the paper, I revised it once again and tried to make a better claim for novelty.
Especially in response 1, that on board a small USV can be mounted light sensors not used generally
(satellite compass). In literature many measurements was provided on cars using marine satellite
compass.
First answers (round 1) to the reviewers’ comments are in red color and second (round 2) in blue in
this document.
Response 1
- It is needed to elaborate more in the added value of the paper related with existing literature
describing the weaknesses identified
Advanced equipment such as MBES, LiDAR, RADAR, cameras, side scan sonars (SSS) are used in bigger
USVs and sounding vessels, equipped with GPS compass for heading determination. Small USV use
basic hydrographic sensors: SBES for depth measurement and GNSS for positioning. For automatic
navigation use internal electronic compass. The article presents results of heading determination using
sensors, which can be mounted on board small USV.
Response 2:
- Specify why particular devices are selected for performing trials related with similar existing in the
market
Selection of used devices are described in Introduction (lines 54-64) because of:
1. Possibilities of mounting them on board the USV (weight and dimensions), only small and light
devices could be used
2. Majority of published measurements’ results of (marine) satellite compasses are realised in
land navigation, on board cars in urban conditions. The article presents results ov
determination of heading in marine navigation.
3. Fluxgate compass used in tests is old and still working.
4. Good solution can be used FOG (mentioned in Conclusions), but is very expensive and was nor
available to rent for tests.
The response is in lines 85-90 in the manuscript
Response 3:
- In the methodology of comparison measurement describe the reason for selection the protocol and
if they have consulted some similar measurements
Selected protocols: HDT, HDM give an information about heading of the USV. COG is a moving
parameter related to moved (GNSS antenna) position. These same or similar parameters are measured
by different devices. It depends of sounding area, observation (GNSS) conditions and magnetic
disturbances.
Please note, used devices give different direction parameters and they can be compared during the
research. There is not possible to select one of the parameter determined by a selected device from
one determined parameter. As an answer, there is not possible to obtain more than one parameter as
HDT or HDM from one sensor.
Additionally, sampling frequency has been added in Table 2.
Response 4:
- Erase/translate the first paragraph of 3.4 Circulation section which is in Polish Thank you for your
suggestion, it was corrected after submission, but the Receiver received a not corrected manuscript.
Response 5
- Describe what are the limitations of your measurement
Satellite compass, as GNSS sensor, works in an open area, so can be impossible to use it in the restricted
area with high port infrastructure, close to high buildings and high moored vessels. Measurement are
provided. GNSS receiver in RTK/RTN mode can be used in the GSM range (operational zone), but the
article is dedicated to hydrographic surveys in coastal areas using small USV with availability of
RTK/RTN corrections. It is partial answer to next question about measurements in open sea.
Response 6:
- Elaborate more in hydro-meteorological factor on the vessel and potential of measurement in
another area (open sea???)
Measurements presented in the manuscript present comparison of used devices for determination
heading/course in a marina, with no wind and waves. Many of my soundings are realised outside the
marina’s breakwater close to public beach. It is open area, but can be called “open sea”. These
measurements can be continued on board a vessel using another, not used on USV devices.
A short explanation has been added in section 2.1:
However parameters measured by the fluxgate and satellite compasses are connected with the
sounding vessel’s hull, COG is determined on the basis of GNSS antenna’s positions’ coordinates. There
is possible difference between COG and parameters measured by two other sensors, which can be
observed in Figure 2. Generally, the hydro-meteorological conditions: wind and wave motion do not
influence on the measuring objects and determined parameters. Discrepancy between HDT and HDM,
and COG is a result of difference between parameters related to water and the ground.
Response 7:
- Generalize your conclusion and propose future research
One paragraph in Conclusion section has been deleted to generalize conclusions.

Reviewer 2 Report
In this paper, results of dynamic tests of three heading meters: electronic and satellite compasses, and GNSS determining COG are presented. These sensors were mounted on USV and compared in different motion patterns of moving straight, performing quick turns and circulations. This paper still needs revision to make a case for novelty
A few suggestions to the authors are as follows:
50-53 Delete “On big sounding vessels, equipped with gyro-com-passes, the gyro-compass heading is used to present the motion vector on water surface. It is also possible to use the satellite compass in good observation conditions, i.e. on open water”
156 replace “staellite” by “satellite”
168 what is “0,0658”and “2,835”. Make the sense clear.
178 what is the frequency of messages transmission (fluxgare) in static and kinematic measurements?
193-216 what is the navigational condition of “3.4 circulations” and “3.3 Quick heading’s changes on reciprocal one”
256 “impacts of hydro-meteorological factors”, was not tested, so reframe the sentence to avoid any confusion.
Author Response
Response to Reviewer 2 Comments
I have carefully considered your comments and I found the feedback to be insightful and valuable. I thank you and the reviewers for the efforts and believe that I have a much-improved version of my paper. The answers to the reviewers’ comments are in red color in this document.
In this paper, results of dynamic tests of three heading meters: electronic and satellite compasses, and GNSS determining COG are presented. These sensors were mounted on USV and compared in different motion patterns of moving straight, performing quick turns and circulations. This paper still needs revision to make a case for novelty
Response 1:
50-53 Delete “On big sounding vessels, equipped with gyro-com-passes, the gyro-compass heading is used to present the motion vector on water surface. It is also possible to use the satellite compass in good observation conditions, i.e. on open water”
In my opinion, the sentence is correct. After deleting it, the paragraph have to be reconsidered because of only one sentence.
Response 2:
156 replace “staellite” by “satellite”
Thank you very much, I has been corrected.
Response 3:
168 what is “0,0658”and “2,835”. Make the sense clear.
Thank you very much, I has been corrected. Commas were replaced by dots.
Response 4:
178 what is the frequency of messages transmission (fluxgare) in static and kinematic measurements?
The frequency of messages transmission (fluxgare) in static and kinematic is 1Hz. It has been mentioned
Response 5:
193-216 what is the navigational condition of “3.4 circulations” and “3.3 Quick heading’s changes on reciprocal one”
Measurements presented in the manuscript present comparison of used devices for determination heading/course in a marina, with no wind and waves. Many of my soundings are realised outside the marina’s breakwater close to public beach. It is open area, but can be called “open sea”. These measurements can be continued on board a vessel using another, not used on USV devices.
Response 6:
256 “impacts of hydro-meteorological factors”, was not tested, so reframe the sentence to avoid any confusion.
Please note, it can be tested in future, but much more on a sounding vessel than determination the course.

Round 2
Reviewer 2 Report
no more suggestions